# Can lung ultrasound score accurately predict the need for surfactant replacement in preterm neonates? A systematic review and meta-analysis protocol

**Letizia Capasso**[1]*, **Daniela Pacella**[2], **Fiorella Migliaro**[1], **Daniele De Luca**[3], **Francesco Raimondi**[1]

**1** Division of Neonatology, Department of Translational Medical Sciences, University Federico II, Naples, Italy, **2** Department of Public Health, University Federico II, Naples, Italy, **3** Service de Pediatrie et Reanimation Neonatale Hopital "A. Beclere" – GHU Paris Saclay, APHP, Paris, France

* letizia.capasso@gmail.com

## Abstract

Respiratory distress syndrome (RDS) is a leading cause of morbidity and mortality in preterm infants due to primary surfactant deficiency. Surfactant replacement has greatly improved the short and long term prognosis of RDS but its administration criteria remain uncertain. Lung ultrasound has been recently shown as a non-invasive, repeatable, bedside tool to estimate parenchymal aeration using a semiquantitative score (LUS). The objective of this systematic review and meta-analysis is to evaluate the accuracy of LUS, assessed on the first day of life, to predict surfactant replacement. Methods will follow the Preferred Reporting Items for Systematic Review and Meta-Analysis Protocols (PRISMA-P) guidelines and the protocol has been registered in PROSPERO database (registration number: CRD42021247888). Primary outcome: in a population of preterm infants, LUS will be compared in neonates who received surfactant replacement versus those who did not. Secondary outcome will be the accuracy of lung ultrasound score to predict the need for $\geq 2$ doses of surfactant.

## Introduction

Respiratory distress syndrome (RDS) is a leading cause of morbidity and mortality in preterm infants due to primary surfactant deficiency. Surfactant replacement has greatly improved the short and long term prognosis of moderate to severe RDS, especially when administered in the first 3 hours of life [1]. Since not all preterm babies with RDS need surfactant, its administration criteria are still being debated.

According to European guidelines, surfactant has to be administrated when oxygen requirement increases above 30% [2]. Rather than being a true index of surfactant deficiency, the oxygen requirement threshold is a proxy that also depends on the respiratory support delivered to the neonate (e.g. the PEEP level) and on the saturation target. These interdependencies may delay surfactant treatment or lead to unnecessary replacement. Lung ultrasound

**Funding:** The authors received no specific funding for this work.

**Competing interests:** The authors have declared that no competing interests exist.

has been recently shown as a non-invasive, repeatable, bedside tool to estimate parenchymal aeration using a semiquantitative score [3,4]. Several groups have shown that a lung ultrasound score (LUS) is a reliable marker to predict the failure of non-invasive support in infants with RDS [5–8]. As a natural consequence, clinicians are investigating LUS thresholds for surfactant replacement as an alternative to or in association with oxygenation markers [3]. Since results may also depend on study populations, score threshold and scoring system, the need for recapitulation and standardization arises. The goal is to establish the role of LUS as a clinically relevant tool for surfactant administration as already hypothesized in the 2019 European guidelines on RDS [2].

## Objective

This systematic review and meta-analysis concerns the accuracy of LUS, assessed on the first day of life, to predict surfactant replacement in preterm neonates.

## Methods

We will follow the Preferred Reporting Items for Systematic Review and Meta-Analysis Protocols (PRISMA-P) guidelines [9] and the protocol has been registered in PROSPERO database (registration number: CRD42021247888). The PRISMA-P 2015 Checklist has been added as supporting information (S1 Checklist).

### Eligibility criteria

The PICOS elements form the basis of clinical questions for this review. Studies will be selected according to the criteria outlined below.

### Participants

Inclusion criteria: all preterm neonates having LUS assessed on the first day of life. Exclusion criteria: neonates with congenital malformations, TORCH infections.

### Interventions

1. lung ultrasound score assessed in preterm neonates on the first day of life before surfactant treatment

2. lung ultrasound scan performed on at least three areas for each lung.

3. we considered studies with the scoring systems including

score = 0 for normal lung imaging (A lines and pleural sliding present);
    score = 1 for alveolar interstitial pattern (B lines not coalescent);
    score = 2 for severe alveolar interstitial pattern (multiple and or coalescent B lines with or without consolidations limited to subpleural space);
    score = 3 for extended consolidation [4,10].

### Comparators

Lung ultrasound score will be compared in preterm neonates who received surfactant treatment versus preterm infants who did not receive it.

Other relevant clinical variables: gestational age (GA); Oxygen Saturation over Inspired Oxygen Fraction (S/F); small for gestational age (SGA); male gender and prenatal steroids administration.

## Outcomes

**Ac**curacy of lung ultrasound score, performed on the first day of life, to predict the subsequently need of surfactant treatment.

## Study design

We will consider any kind of trial on the LUS accuracy to predict need of surfactant treatment in preterm neonates written in English between 2011 to 2021.

## Information sources and search strategy

The databases Pubmed, Scopus, Biomed Central, Cochrane library will be consulted from January 2011 to June 2021.

We believe that this 10 years interval is appropriate to give the reader complete information. The original approach to neonatal lung ultrasound was purely descriptive and no score strategy was contemplated. Our group published an image classification system in 2012 [7] but Brat and coworkers adopted the numerical score for the first time in 2015 that is currently widely used [4,10].

The key words to be used are: lung ultrasound and surfactant and neonate. Also, we will scan the reference lists of included studies identified through the search.

## Data management

Literature search's results will be shared among all the authors to approve the eligibility of selected studies according to the eligibility criteria. All authors will pay attention to ascertain duplicate publications or multiple reports of the same study.

## Selection process

Two reviewers will independently select eligible abstracts and verify the acceptability of the full studies. Two author will extract the data. Two independent authors will assess the risk of bias in each individual study as well as assess the possible publication bias. Results will be compared and discussed among all the authors.

## Data collection process and items

Information will be extracted using a standardized form and reported in a Microsoft Excel (Redmond, WA: Microsoft, 2013) spreadsheet.

The following data will be extracted from the studies:

author, year of publication, number of neonates included, number of areas scored, LUS value predictive for surfactant treatment and AUC; reported (or derived) raw true negative, false negative, true positive and false positive will be extracted to compute Sensitivity, Specificity, Positive Predictive Value, Negative Predictive Value, positive and negative Likehood ratios, diagnostic Odds Ratio.

Clinical characteristics for treated and control subjects to be recorded will be: median of gestational age; S/F ratio; percentage of male infants, SGA and use of prenatal steroids.

Also will be reported the need for mechanical ventilation, occurrence of pneumothorax, bronchopulmonary dysplasia and death for treated and control subjects.

## Outcomes and prioritisation

Our primary outcome will be to test the accuracy of LUS performed in the first day of life to predict surfactant treatment in preterm neonates.

Secondary outcomes will be: accuracy of LUS to predict the need for $\geq 2$ doses of surfactant in preterm neonates.

## Risk of bias in included studies

Quality and risk of bias for the systematic review and meta-analysis will be assessed using an adapted version of the Cochrane's tool for Risk Of Bias in Non-randomized Studies (ROBINS-1). All eligible studies will be considered for the meta-analysis, regardless of their quality and assessed risk of bias. However, sensitivity analysis will be conducted excluding studies with high risk of bias. Regarding the risk of meta-biases, this is discussed and detailed in the corresponding section.

## Data synthesis

Pooled estimates of sensitivity, specificity and DOR will be provided for the prediction of the first and second surfactant dose. Given that the meta-analysis will include studies with different LUS thresholds due to the lack of standardization, high heterogeneity is expected. Heterogeneity will be assessed using both I-square statistics and Kendall's tau. For high heterogeneity studies, or for studies which involve populations with different baseline clinical and demographic characteristics, subgroup analysis will be conducted. Meta-regression may also be employed, as appropriate. Additionally, summary receiver operator characteristic (sROC) curve will be computed as well as the area under the curve (AUC).

## Meta-biases

Authors will assess each study's sampling strategy, representativeness and comparability of the samples and use of comparable instruments. Along with of risk of bias in individual studies, biases in the meta-analysis will be assessed as follows:

- risk of publication bias will be assessed with the visual inspection of funnel plots and of the computed sROCs;

- Studies with high suspicion of selective reporting bias (SRB) will be processed as follows:

  1. if applicable, authors will be contacted to integrate unclear or missing observations, data or outcomes;

  2. studies will be excluded from the computation of the pooled estimates.

## Confidence in cumulative evidence

The strength of the body of evidence will be assessed as follows:

- For the interpretation of the study contribution to the findings, references will be made to the prior assessment of methodological quality of the included studies;

- Consistency and inconsistency across findings will be assessed and any incoherent or contradictory evidence will be highlighted and discussed;

- GRADE or CERQual approaches may be employed for standardized assessment of cumulative evidence quality.

## Supporting information

**S1 Checklist. PRISMA-P 2015 checklist.**
(DOCX)

## Author Contributions

**Conceptualization:** Letizia Capasso, Daniele De Luca, Francesco Raimondi.

**Data curation:** Letizia Capasso, Daniela Pacella, Fiorella Migliaro, Francesco Raimondi.

**Formal analysis:** Letizia Capasso, Daniela Pacella.

**Methodology:** Daniela Pacella.

**Supervision:** Daniele De Luca, Francesco Raimondi.

**Validation:** Daniela Pacella.

**Writing – original draft:** Letizia Capasso, Daniela Pacella.

**Writing – review & editing:** Daniele De Luca, Francesco Raimondi.

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
