## [Decision Letter · Decision Letter 0]

1 Jun 2021

PONE-D-21-12565

Can lung ultrasound score accurately predict the need for surfactant replacement in preterm neonates? A systematic review and meta-analysis protocol.

PLOS ONE

Dear Dr. capasso,

Thank you for submitting your manuscript to PLOS ONE. After careful consideration, we feel that it has merit but does not fully meet PLOS ONE’s publication criteria as it currently stands. Therefore, we invite you to submit a revised version of the manuscript that addresses the points raised during the review process.

We look forward to receiving your revised manuscript.

Kind regards,

Kazumichi Fujioka

Academic Editor

PLOS ONE

Journal Requirements:

2. Please confirm that you have included all items recommended in the PRISMA-P checklist including:

- the full electronic search strategy used to identify studies with all search terms and limits for at least one database.

- an explanation for why the search inclusion dates begin in 2011

- an explanation of how risk of bias will be assessed

Thank you.

Reviewers' comments:

Reviewer's Responses to Questions

**Comments to the Author**

1. Does the manuscript provide a valid rationale for the proposed study, with clearly identified and justified research questions?

Reviewer #1: Yes

Reviewer #2: Yes

2. Is the protocol technically sound and planned in a manner that will lead to a meaningful outcome and allow testing the stated hypotheses?

Reviewer #1: Yes

Reviewer #2: Partly

3. Is the methodology feasible and described in sufficient detail to allow the work to be replicable?

Reviewer #1: Yes

Reviewer #2: Yes

4. Have the authors described where all data underlying the findings will be made available when the study is complete?

Reviewer #1: Yes

Reviewer #2: Yes

5. Is the manuscript presented in an intelligible fashion and written in standard English?

Reviewer #1: Yes

Reviewer #2: Yes

6. Review Comments to the Author

You may also provide optional suggestions and comments to authors that they might find helpful in planning their study.

Reviewer #1: This manuscript reports a systematic review and meta-analysis concerns the accuracy of LUS, assessed on the first day 52 of life, to predict surfactant replacement. This is a good paper that increases the existing scientific knowledge in this compound.

Reviewer #2: Thank you for giving me the opportunity to review this Study Protocol by Capasso L. et al. titled « Can lung ultrasound score accurately predict the need for surfactant replacement in preterm neonates? A systematic review and meta-analysis protocol."».

Overall the protocol is useful and accurate. The objectives are clearly stated, the search strategy and the statistical methodology are very well detailed, making it possible to decide on the level of proof about the studied technique with maximum bias avoidance.

I have a few comments/suggestions:

- The authors aim to study the accuracy of a referenced ultrasound score, LUS (Brat et al., JAMA Pediatr 2015), to predict surfactant treatment in premature infants. Is there not a way to include other studies done with comparable objective and technique, but assessing parenchymal disease by the means of other scores ?

- The need for invasive mechanical ventilation, the occurrence of pneumothorax and air leaks, of bronchopulmonary dysplasia and death could be studied as secondary endpoints. These elements would provide some benchmarks for future randomized controlled trials comparing ultrasounds-guided surfactant replacement to conventional modalities, as oxygenation index or fraction of inspired oxygen.

- Statistical analysis should mention term subgroup analyzes.

- The authors should please comment on the contributions of their study protocol compared to the article "Neonatal lung ultrasonography to evaluate need for surfactant or mechanical ventilation: a systematic review and meta-analysis.", published in Arch Dis Child Fetal Neonatal Ed. 2020 Mar;105(2):164-171. doi: 10.1136/archdischild-2019-316832.

Best regards

7. PLOS authors have the option to publish the peer review history of their article (what does this mean?). If published, this will include your full peer review and any attached files.

Reviewer #1: No

Reviewer #2: No

---

## [Author Response · Author response to Decision Letter 0]

10 Jul 2021

To: Dr Kazumichi Fujioka

Academic Editor 

PLOS ONE Naples 10/07/2021 

Thank you for the interest in our protocol. 

We now provide a revised version of the manuscript following your observations and the reviewers’ suggestions. Below you will find a point by point answer to the comments.

EDITOR’S COMMENTS

1. Please ensure that your manuscript meets PLOS ONE's style requirements, including those for file naming. The PLOS ONE style templates can be found at…….

Answer: the renewed manuscript strictly adheres to the journal’s style requirements

2. Please confirm that you have included all items recommended in the PRISMA-P checklist including:

- the full electronic search strategy used to identify studies with all search terms and limits for at least one database.

- an explanation for why the search inclusion dates begin in 2011

- an explanation of how risk of bias will be assessed

Answer: we closely follow all the PRISMA-P checklist. 

Moreover:

-A search strategy has been now detailed from lines 88 to 96. 

-The original approach to neonatal lung ultrasound was purely descriptive and no score strategy was contemplated. We published an image classification system in 2012, subsequently, Brat and coworkers adopted the numerical score that is currently widely used. We believe that a 10 years interval is appropriate to give the reader complete information and we have extended our search to June 2021 (line 86).

-Risk of bias assessment is described in the section “Risk of bias in included studies”. Changes have been made to the section to make it clearer for the reader (line 123-128).

-We added the PROSPERO registration number (lines 25 and 56) in the Abstract and Introduction sections.

 3.………..We note that you have stated that you will provide repository information for your data at acceptance. Should your manuscript be accepted for publication, we will hold it until you provide the relevant accession numbers or DOIs necessary to access your data. If you wish to make changes to your Data Availability statement, please describe these changes in your cover letter and we will update your Data Availability statement to reflect the information you provide”

Answer: we would change the Data Availability statement in: Data available on request.

Answer: we added caption at the end of manuscript (line 160) and updated citation in text (line 58-59)

REVIEWERS' COMMENTS

Reviewer #1

This manuscript reports a systematic review and meta-analysis concerns the accuracy of LUS, assessed on the first day of life, to predict surfactant replacement. This is a good paper that increases the existing scientific knowledge in this compound.

Reviewer #2

Thank you for giving me the opportunity to review this Study Protocol by Capasso L. et al. titled « Can lung ultrasound score accurately predict the need for surfactant replacement in preterm neonates? A systematic review and meta-analysis protocol."».

Overall the protocol is useful and accurate. The objectives are clearly stated, the search strategy and the statistical methodology are very well detailed, making it possible to decide on the level of proof about the studied technique with maximum bias avoidance.

I have a few comments/suggestions:

1. The authors aim to study the accuracy of a referenced ultrasound score, LUS (Brat et al., JAMA Pediatr 2015), to predict surfactant treatment in premature infants. Is there not a way to include other studies done with comparable objective and technique, but assessing parenchymal disease by the means of other scores ?

Answer: Brat et al applied to neonates with RDS a numerical score from the adult literature (Brat et al., JAMA Pediatr 2015) that has been validated versus a number of physiological variables. 

This physiological correlation represents a point of strength of this non invasive tool and probably stands behind its relative popularity.

This is less true for other scores where either a formal validation is lacking or a qualitative (i.e. type 1, type 2, type 3) rather than a semiquantitative strategy was adopted . 

A direct comparison, therefore, appears inappropriate.

2. The need for invasive mechanical ventilation, the occurrence of pneumothorax and air leaks, of bronchopulmonary dysplasia and death could be studied as secondary endpoints. These elements would provide some benchmarks for future randomized controlled trials comparing ultrasounds-guided surfactant replacement to conventional modalities, as oxygenation index or fraction of inspired oxygen.

Answer: the reviewer has rightly underlined relevant clinical information that will now be acquired ( cfr line 112-117).

3. Statistical analysis should mention term subgroup analyzes.

Answer: Subgroup analysis was considered as a method to be employed especially in case high heterogeneity is detected among the eligible studies, but also in the case of populations with different characteristics. The section has also been slightly rephrased to evidence this aspect and make it clearer to the reader (lines 133 – 135). 

4. The authors should please comment on the contributions of their study protocol compared to the article "Neonatal lung ultrasonography to evaluate need for surfactant or mechanical ventilation: a systematic review and meta-analysis.", published in Arch Dis Child Fetal Neonatal Ed. 2020 Mar;105(2):164-171. doi: 10.1136/archdischild-2019-316832.

Answer: the mentioned manuscript has the invaluable credit to be the first review and metanalysis on the topic. The authors included the three single center studies available in 2018 where a wide interval of gestational age (term and preterm) and different causes of respiratory distress (e.g. RDS and TTN) were included.

Since then, a few pertinent manuscripts have been published, more investigation groups have been involved and the number of babies has substantially increased. 

The aim of our protocol differs from Razak et al in that it is focused only on preterm neonates with RDS that need surfactant replacement (line 51-52).

---

## [Editor Report · Decision Letter 1]

15 Jul 2021

Can lung ultrasound score accurately predict the need for surfactant replacement in preterm neonates? A systematic review and meta-analysis protocol.

PONE-D-21-12565R1

Dear Dr. capasso,

We’re pleased to inform you that your manuscript has been judged scientifically suitable for publication and will be formally accepted for publication once it meets all outstanding technical requirements.

Kind regards,

Kazumichi Fujioka

Academic Editor

PLOS ONE
---

## [Editor Report · Acceptance letter]

19 Jul 2021

PONE-D-21-12565R1 

Can lung ultrasound score accurately predict the need for surfactant  replacement in preterm neonates? A systematic review and meta-analysis protocol. 

Dear Dr. Capasso :

I'm pleased to inform you that your manuscript has been deemed suitable for publication in PLOS ONE. Congratulations! Your manuscript is now with our production department. 

Kind regards, 

on behalf of

Dr. Kazumichi Fujioka 

Academic Editor

PLOS ONE